# SparRL: Graph Sparsification via Deep Reinforcement Learning

## Abstract

Graph sparsification concerns data reduction where an edge-reduced graph of a similar structure is preferred. Existing methods are mostly sampling-based, which introduce high computation complexity in general and lack of flexibility for a different reduction objective. We present SparRL, the first general and effective reinforcement learning-based framework for graph sparsification. SparRL can easily adapt to different reduction goals and promise graph-size-independent complexity. Extensive experiments show that SparRL outperforms all prevailing sparsification methods in producing high-quality sparsified graphs concerning a variety of objectives. As graph representations are very versatile, SparRL carries the potential for a broad impact.

## 1 Introduction

Graphs are the natural abstraction of complex correlations found in numerous application domains including social media, communications, transportation, and medicine discovery, and thus have been studied in many scientific disciplines such as computer science, mathematics, engineering, sociology, and economics. The interconnectedness of a graph can be leveraged to explore, infer, and learn latent structures as well as global and local information embedded in the graph. However, modern graphs can be very complex, causing their analysis, computing, and utilization inefficient. This has motivated the study of *graph sparsification* in the past two decades (Batson et al., 2013; Teng, 2016), where the objective is to prune edges from a graph to produce a sparsified graph while preserving user-defined metrics in query evaluation or knowledge inferring. The most commonly adopted metrics include the graph spectrum and the effective resistance of edges (Spielman & Srivastava, 2011; Spielman & Teng, 2011). Sparsification techniques developed w.r.t these metrics have been applied to domains such as power grid management (Zhao et al., 2014; Zhao & Feng, 2017), integrated circuit simulation (Zhao et al., 2015), and influence maximization (Shen et al., 2017; Mathioudakis et al., 2011).

So far, many graph sparsification techniques are sampling-based (Fung et al., 2019). While effective, they all introduce high computation complexity and lack the flexibility to preserve different graph properties in many applications, e.g., approximate graph analysis (Iyer et al., 2018; Ahn et al., 2012; Satuluri et al., 2011a; Zhao, 2015), privacy preserving (Upadhyay, 2013; Arora & Upadhyay, 2019), and representation learning (Calandriello et al., 2018). Thus, a general, flexible graph sparsification technique for various reduction objectives and application domains is highly desired.

We present SparRL, a general graph sparsification framework empowered by deep reinforcement learning that can be applied to any edge sparsification task with a customized reduction goal. Consider the example shown in Figure 1, by setting modularity preservation as the edge reduction objective function, SparRL can prune a user-defined amount of edges from the original graph and still preserve the substructure modularity. To improve learning efficiency and convergence rate of SparRL, we initialize the initial state by randomly sparsifying the graph before each training episode, use Double DQN (Van Hasselt et al., 2016) with Prioritized Replay (Schaul et al., 2015), and employ $\epsilon$-greedy exploration for searching for the optimal pruning strategy.

We test SparRL using a wide range of graph datasets and metrics including PageRank, community structure, and pairwise shortest-path distance. As a result, SparRL outperforms all baselines on preserving PageRank, community structure, and single-pair shortest path (SPSP) preservation at a variety of edge-kept ratios.

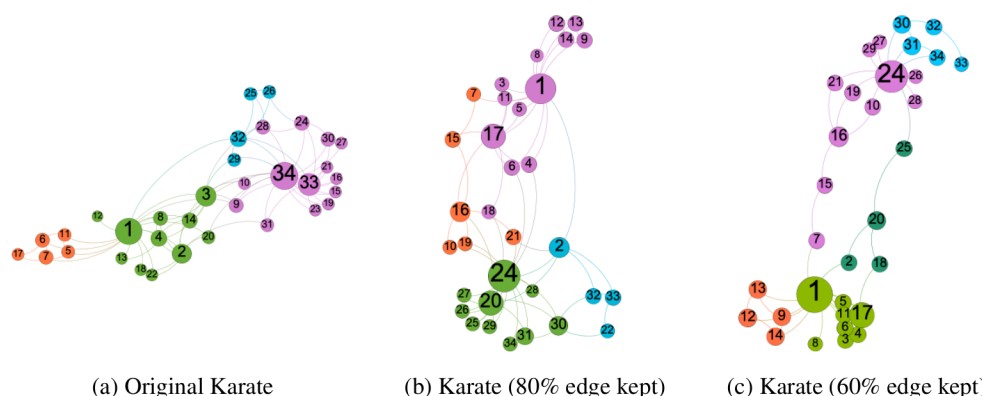

(a) Original Karate        (b) Karate (80% edge kept)        (c) Karate (60% edge kept)

Figure 1: Applying SparRL to Zachary's karate club (Zachary, 1977) graph by setting modularity preservation the edge reduction goal. Different colors denote different modularity-based partitions, while a node size scales with its degree. Modularity partition computed and plotted with Gephi (Bastian et al., 2009)

In summary, the contributions of SparRL are the following:

- A novel reinforcement learning-based general-purpose graph sparsification framework with modular, task-adaptive, and easily modifiable components;
- Task flexibility because of the plug-in reward function;
- Task scalability as SparRL's time complexity is independent from the size of a graph; and
- Simple to use time v.s. performance trade-offs.

The source code of SparRL can be found at `https://anonymous.4open.science/r/SparRL-PyTorch-F545/`.

## 2 RELATED WORK

Originated from the cut problem (Benczúr & Karger, 1996), graph sparsification has drawn extensive research interests (Batson et al., 2012; Lee & Sun, 2015; 2017; Spielman & Srivastava, 2011; Spielman & Teng, 2011). A straightforward approach to this problem is to remove edges from a graph with probabilities proportional to the edge weights. However, this approach may fail for a highly structured graph, e.g., a graph with a small cut value where the sampling may exclude the cut edges, thus losing the connectivity of the original graph. An improved approach is to use the k-neighbors sparsifier (Sadhanala et al., 2016), which performs local sparsification node-by-node via retaining edges of nodes that have degree smaller than a predefined threshold $\tau$ and removing edges of nodes (proportionally to their weights) that have degree greater than $\tau$. Another method is to remove edges proportionally to their effective resistance, which measures the importance of an edge in preserving the minimum distance between two nodes. This approach will result in only essential edges (on the order of $O(n\,\mathrm{polylog}\,n)$) being kept. Although there have been advancement in fast approximation of effective resistance (Koutis et al., 2012; 2016), the graph size remains a dominating factor in its computing complexity.

A related line of research in graph theory, namely graph spanner (Peleg & Ullman, 1989), aims to compute a subgraph that satisfies certain reachability or distance constraints, e.g., $t$-spanner (Fekete & Kremer, 2001; Dragan et al., 2011), where the geometric distance of every vertices pair in the subgraph is at most $t$ times of the real distance in the original graph. Therefore, $t$ is also named the stretch parameter, which needs to be specified for a spanner algorithm. However, spanner has no guarantee on the edge reduction ratio. In contrast, SparRL takes the edge reduction ratio as input and make edge pruning decision via a learned model aiming to best preserve the desired graph structural property. Moreover, techniques developed for the spanner problem are subject to geometric distance preservation, which make them hard to be generalized for the wide spectrum of graph sparsification objectives. More details on graph sparsification techniques can be found in survey papers (Batson et al., 2013; Teng, 2016).

In recent years, learning-based algorithms have gained popularity as they allow for direct optimization on a task's objective and have demonstrated superior performance compared to traditional methods.

Surprisingly, little work has applied learning to graph sparsification. GSGAN (Wu & Chen, 2020) approaches graph sparsification using GAN (Goodfellow et al., 2014), whose goal is to preserve the community structure of a graph by learning to generate a new graph. While effective, GSGAN could introduce edges that are not in the original graph, thus compromising the graph sparsification objective on many real-world networks where establishing new edges/connections is resource-intensive (e.g., road network). RNet-DQN (Darvariu et al., 2020) uses RL to process graphs, where the focus is to enhance the resilience of graph via adding edge instead of sparsification. GDPNet (Wang et al., 2019) also uses RL but the goal is graph representation learning instead of graph sparsification. The most relevant study to ours is NeuralSparse (Zheng et al., 2020), whose focus again is representation learning. The subgraphs are produced via supervised learning and need to go through graph neural networks for downstream classification. Due to these constraints, classical analytic benchmarks such as community detection or shortest path computing from the traditional graph sparsification studies are missing in (Zheng et al., 2020). In comparison, SparRL outputs a sparsified graph where existing graph analytic algorithms can be directly applied. To the best of our knowledge, SparRL is the first task-adaptive and effective RL-based framework for graph sparsification.

## 3  PRELIMINARIES

Consider a $T$-step episodic task. At each timestep $t \in [1, T]$, the RL agent uses its policy $\pi_\theta(a_t|s_t)$ to choose action $a_t$ based on state $s_t$ from the environment. Then, the environment responds with reward $r_t$ and next state $s_{t+1}$. This sequential decision-making process is formulated as a Partially-Observable Markov Decision Process (POMDP) defined by the tuple $(\mathcal{S}, \mathcal{A}, P, R, \Omega, \mathcal{O}, \gamma)$, where $\mathcal{S}$ is the state space, $\mathcal{A}$ is the action space, $P(s'|s,a)$ is the transition probability function, $R$ is the reward function, $\Omega$ is the set of observations, $\mathcal{O}$ is the observation probability function, and $\gamma \in [0, 1]$ is the discount factor. The objective of the RL agent is to find $\pi_\theta$ that can maximize the sum of discounted rewards, or the *return* $R_t = \sum_{i=t}^{T} \gamma^{i-t} r_i$.

Related to (PO)MDP is the concept of Q-function $Q^\pi : \mathcal{S} \times \mathcal{A} \to \mathbb{R}$, which describes the total expected reward by taking action $a$ in state $s$ and then following policy $\pi$ thereafter. The element that we want to obtain is the optimal Q-function $Q^*$, from which we can trivially derive the optimal policy $\pi^*(s) = argmax_a Q^*(s, a)$. One way to learn $Q^*$ is through Q-learning (Watkins, 1989):

$$Q_{t+1}(s_t, a_t) \leftarrow Q_t(s_t, a_t) + \alpha_t \left[ r_{t+1} + \gamma max_{a'} Q_t(s_{t+1}, a') - Q_t(s_t, a_t) \right]$$

, where $\alpha_t \in (0, 1]$ is the step size. Accompanying the rise of deep learning, many variations of Q-learning have been developed, among which Deep Q-Network (DQN) (Mnih et al., 2015) has gained popularity due to its success in playing Atari games. There exist several major improvements of DQN, including Double DQN (Van Hasselt et al., 2016), Prioritized Replay (Schaul et al., 2015), and Dueling DQN (Wang et al., 2016). In this work, we choose Double DQN and Prioritized Replay as a component of SparRL since it has significant improvement over DQN with small code changes.

## 4  SPARRL FRAMEWORK

In this section, we first provide an overview of our approach, then detail its components and design rationale.

### 4.1  FRAMEWORK OVERVIEW

The overall goal is to find an edge sparsified graph $G' = (V'.E')$ that approximates the original graph $G = (V, E)$ measured over some performance metric.

We treat this as an episodic task, where edges are sequentially pruned from $G$. Each timestep, a subgraph of edges are sampled from $E'$ and SparRL's action consists of choosing an edge to prune from the sampled batch of edges. This continues until $T$ edges are pruned from the graph which will produce $G'$. We describe this process in Algorithm 1.

We exploit the simplicity of the environment, by allowing the initial state $s_0$ to be sampled from any state in the state space $\mathcal{S}$. This is implemented as a preprocessing step, where before each episode,

---

**Algorithm 1:** SparRL Framework

**for** $i = 0$ **to** $num\_episodes$ **do**
  $G' \leftarrow$ clone $G$
  Sample $T \sim \mathcal{U}(1, T_{max})$
  Sample $p \sim \mathcal{U}(0, 1)$
  $T_p = min(|E| * p, |E| - T)$
  Randomly prune $T_p$ edges from $G'$
  **for** $t = 0$ **to** $T$ **do**
    $H_t \leftarrow$ Randomly sample $|H|$ edges from $G'$
    $d_{H_t} \leftarrow$ Degrees of nodes in $H_t$
    $q\_values \leftarrow f_{SparRL}(H_t, d_{H_t}, \frac{|E_{G'}|}{|E_G|})$
    $a_t \leftarrow$ Sample action using $\epsilon$-greedy exploration from $q\_values$
    Prune edge $a_t$ from $G'$
    $r_t \leftarrow R(G')$
    Save timestep trajectory $(s_t, s_{t+1}, a_t, r_t)$ in prioritized replay buffer
    Train $f_{SparRL}$ on batch of trajectories
  **end for**
**end for**

---

we randomly prune $T_p$ edges from $G$ to produce the initial sparsified graph at the first timestep, $G'_0$. The percentage of edges to prune is sampled from a continuous uniform distribution $p \sim \mathcal{U}(0, 1)$ and we set $T_p = min(|E| * p, |E| - T)$. This allows for SparRL agent to start in any part of the state space, without having to initially prune $T_p$ edges to visit it. Thus, efficient exploration of the state space is removed from the behavior of the trained policy.

## 4.2 MDP FORMULATION

We formalize the task of sparsification as an POMDP that is solved using reinforcement learning; however, for ease of explanation, we will simplify it to an MDP. The state $s_t$ includes the subgraph $H_t$, the degrees of all the nodes in the subgraph $d_{H_t}$, and the percentage of edges left in the graph $|E_{G'_t}|/|E_G|$, where $|E_{G'_t}|$ is the number of edges in the sparsified graph and $|E_G|$ is the number of edges in the original graph. The reward $r$ is dependent on the properties of the graph that is encouraged to preserved, and thus will be discussed later in the experiments section. Each timestep we compute the reward after an edge is pruned to avoid sparse rewards. The action $a_t$ is the edge to prune. The probability transition function consists of a randomly sampled edge subgraph from $G'$, where each edge has an equal chance to be sampled. The episode goes on for $T$ timesteps until it is terminated. For training, we keep $T$ small at $T = 8$, as we want to encourage more time on exploration rather than an episodes initialized at a random initial state.

## 4.3 POLICY LEARNING

We use Double DQN (Van Hasselt et al., 2016) to represent the SparRL sparsification policy that is parameterized by a deep neural network. The policy is trained over sampled batch of trajectories, sampled using prioritized replay (Schaul et al., 2015).

The model architecture, shown in Figure 2, is composed of the node encoder, edge encoder, and action-value function head. The node encoder first looks up the initial node embedding for all the nodes in the graph, which are trained jointly with the model. Then, the node encoder uses a GAT (Veličković et al., 2017) that applies self-attention to the neighborhood of each node to produce a new node embedding. Each node embeddings of the subgraph are then separately combined with its degrees, in-degree and out-degree if the graph is directed, and the ratio of edges left in the graph, $|E_{G'_t}|/|E_G|$. The edge encoder combines two nodes that represent an edge. Finally, the action-value head outputs the q-values, $Q(s_t, a_0) \ldots Q(s_t, a_n)$, for each edge in the subgraph. The GAT in the node-encoder uses a single fully-connected layer with 1 unit for computing the attention coefficients, both the main part of the node-encoder and edge-encoder consists of two fully-connected layers with 128 units each that are followed by LeakyReLU activation, and the action-value head consists of a single fully-connected layer with one unit.

Thus, the model approximates the Q-value function:

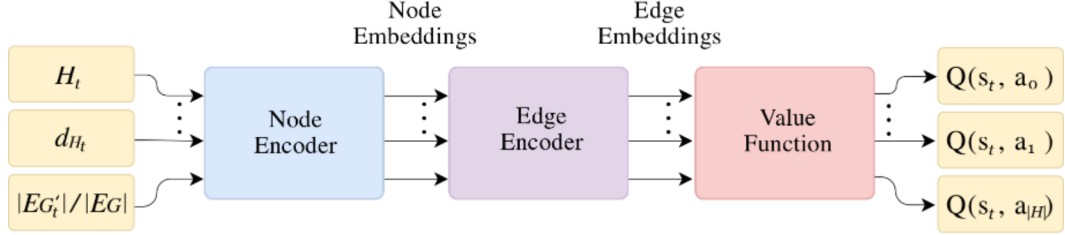

Figure 2: The SparRL model architecture that consists of the node encoder, edge encoder, and action-value head. The input to the model includes the subgraph, the degrees of the node, and ratio of edges still in the graph. The node encoder combine uses a GAT on the 1-hop neighborhood of each node embedding and then combines the output node embedding with its degrees and the ratio of edges left in the graph. The edge encoder will combine each pair of nodes that represent an edge. The action-value function edge will produce the q-value for each edge.

Table 1: Graph datasets used in the experiments.

| Dataset | Twitter | Facebook | YouTube (Top-100 Comm.) | Amazon (Top-500 Comm.) | Email | CiteSeer |
|---|---|---|---|---|---|---|
| $|V|$ | 81,306 | 4,039 | 4,890 | 4,259 | 1,005 | 3,264 |
| $|E|$ | 1,768,149 | 88,234 | 20,787 | 13,474 | 16,064 | 4,536 |

$$f_{SparRL}(H_t, d_{H_t}, \frac{|E_{G'_t}|}{|E_G|}) = Q(s_t, a_0), \ldots, Q(s_t, a_{|H|}). \tag{1}$$

Each edge of the subgraph is independently ran through the network, so the subgraph length $|H|$ is not fixed by the network. Therefore, any number of edges can be considered to be pruned at each timestep during test time.

## 5 EXPERIMENTS

We validate the effectiveness of SparRL using a variety of real-world datasets and test its components performance over several metrics. The key observations include:

- SparRL demonstrates superior performance to existing sparsification methods on representative graph metrics over graphs of different scales;
- SparRL can outperform the $t$-spanner method for preserving Single-Pair-Shortest-Path (SPSP) distances at the same edge kept ratio; and
- SparRL allows for a simple time vs. performance trade-off by modifying $|H|$.

### 5.1 EXPERIMENT SETUP

**Datasets**. We test SparRL using graphs from a variety of domains: Twitter (Leskovec & Mcauley, 2012), Facebook (Leskovec & Mcauley, 2012), YouTube (Mislove et al., 2007), Amazon (Yang & Leskovec, 2015), Email-Eu-Core (Yin et al., 2017), and CiteSeer (Sen et al., 2008). Table 1 summarizes the number of nodes and edges of each graph.

**Baseline Methods**. We compare SparRL with a wide range of sparsification methods: Random Edge (RE), Local Degree (LD) (Hamann et al., 2016), Edge Forest Fire (EFF) (Hamann et al., 2016), Algebraic Distance (AD) (Chen & Safro, 2011), L-Spar (LS) (Satuluri et al., 2011b), Simmelian Backbone (SB) (Nick et al., 2013), and Quadrilateral Simmelian Backbone (QSB) (Nocaj et al., 2014). The details of the baseline methods are provided in Appendix A.1.

**Metrics**. We assess the sparsification methods by examining how well they preserve the topological structure of the original graph w.r.t different metrics, namely PageRank, community structures, and pairwise shortest path distance (Hamann et al., 2016). For each metric, we evaluate the performance of a sparsification method by running the method eight times independently and report the average.

**SparRL Setup**. Next, we detail our hyperparameter values and settings that are associated with the Double DQN algorithm used by our sparsifier agent. We update the target DQN network by apply a

Table 2: Comparison of PageRank preservation via the Spearman's $\rho$ index, where $r$ is the edge kept ratio.

| Method | Twitter | | | | Facebook | | | | Amazon (Top-500 Comm.) | | | |
|---|---|---|---|---|---|---|---|---|---|---|---|---|
| | $r$=0.2 | $r$=0.4 | $r$=0.6 | $r$=0.8 | $r$=0.2 | $r$=0.4 | $r$=0.6 | $r$=0.8 | $r$=0.2 | $r$=0.4 | $r$=0.6 | $r$=0.8 |
| SparRL | **0.846** | **0.944** | **0.984** | **0.995** | **0.942** | **0.982** | **0.996** | **0.998** | **0.779** | **0.907** | **0.944** | **0.988** |
| LD | 0.512 | 0.775 | 0.876 | 0.929 | 0.899 | 0.979 | 0.995 | 0.995 | 0.755 | 0.884 | 0.929 | 0.978 |
| RE | 0.763 | 0.864 | 0.924 | 0.967 | 0.802 | 0.905 | 0.955 | 0.982 | 0.549 | 0.749 | 0.871 | 0.948 |
| EFF | 0.628 | 0.725 | 0.811 | 0.892 | 0.629 | 0.801 | 0.919 | 0.980 | 0.669 | 0.728 | 0.888 | 0.969 |
| AD | 0.520 | 0.690 | 0.837 | 0.940 | 0.408 | 0.519 | 0.637 | 0.782 | 0.230 | 0.341 | 0.553 | 0.769 |
| LS | 0.771 | 0.826 | 0.857 | 0.892 | 0.648 | 0.830 | 0.924 | 0.960 | 0.589 | 0.640 | 0.763 | 0.859 |
| SB | 0.581 | 0.689 | 0.761 | 0.811 | 0.379 | 0.582 | 0.681 | 0.740 | 0.247 | 0.348 | 0.397 | 0.399 |
| QSB | 0.642 | 0.746 | 0.794 | 0.821 | 0.512 | 0.585 | 0.671 | 0.737 | 0.280 | 0.354 | 0.399 | 0.399 |

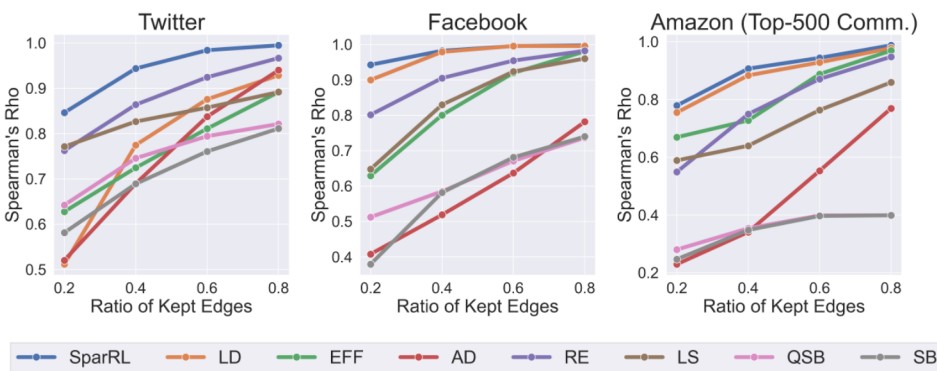

Figure 3: PageRank preservation measures over Spearman's $\rho$. SparRL outperforms all other methods on all cases.

soft updates after training on a batch of trajectories each timestep: $\theta'_{target} \leftarrow \varphi\theta_{target} + (1 - \varphi) * \theta$, where $\theta_{target}$ are the parameters of the target DQN network and $\theta$ are the parameters of the current DQN network policy. We find that $\varphi = 0.001$ corresponds to stable results. We use $\alpha = 0.6$ and $\beta = 0.4$ for Prioritized Replay, $\gamma = 0.95$ for the discounted return, and initially set $\epsilon = 0.99$ for $\epsilon$-greedy exploration and decay it to $0.05$ over the first 10k policy update steps. We keep the learning rate of the model fixed at $0.0002$ during the entire training process.

When training the model for all experiments we set $T_{max} = 8$, where we sample $T$ between $[1, T_{max}]$ before each episode, the maximum prepruning percent $p = 0.8$, and train until negligible improvements are found over pruning $10\%$ of the edges. Training the SparRL networks typically takes 1h–4h for the smaller graphs and roughly 6h–12h for the larger graphs using an Intel Core i9-11900k CPU and RTX 3090 GPU. However, the run-time complexity when evaluating is simply $O(|H| * T)$ as we are predicting over a subgraph of length $|H|$ for a total of $T$ times.

When evaluating on various test datasets, we act greedily w.r.t the learned DQN policy and prune the edge that corresponds to the maximum q-value in the output.

## 5.2 EFFECTIVENESS OF SPARRL

**PageRank Preservation**. PageRank serves a critical centrality metric for many ranking-based graph applications. We examine the sparsification methods by comparing the Spearman's $\rho$ rank correlation coefficient (Myers & Well, 2003) between the PageRank score of the original graph and of the sparsified graph at multiple edge-kept ratios, defined as $|E_{G'}|/|E|$, where $|E_{G'}|$ is the number of edges in the sparsified graph. We define the reward for PageRank as the difference of Spearman's $\rho$ rank correlation coefficients between $G'$ and $G$:

$$r_{pr} = \rho_{G'} - \rho_G,$$

Table 3: Comparison of community structure preservation over the ARI index, where $r$ is the edge kept ratio.

| Method | YouTube (Top-100 Comm.) | | | | Email-Eu-Core | | | | Amazon (Top-500 Comm.) | | | |
|--------|-------|-------|-------|-------|-------|-------|-------|-------|-------|-------|-------|-------|
| | $r$=0.2 | $r$=0.4 | $r$=0.6 | $r$=0.8 | $r$=0.2 | $r$=0.4 | $r$=0.6 | $r$=0.8 | $r$=0.2 | $r$=0.4 | $r$=0.6 | $r$=0.8 |
| SparRL | **0.084** | **0.230** | **0.323** | **0.253** | **0.651** | **0.705** | **0.527** | **0.429** | **0.248** | **0.285** | **0.257** | **0.269** |
| LD | 0.052 | 0.082 | 0.150 | 0.145 | 0.278 | 0.30 | 0.278 | 0.207 | 0.240 | 0.238 | 0.236 | 0.236 |
| RE | 0.048 | 0.144 | 0.157 | 0.152 | 0.226 | 0.318 | 0.348 | 0.312 | 0.141 | 0.224 | 0.236 | 0.235 |
| EFF | 0.029 | 0.092 | 0.117 | 0.134 | 0.407 | 0.385 | 0.347 | 0.334 | 0.109 | 0.180 | 0.223 | 0.232 |
| AD | 0.035 | 0.068 | 0.111 | 0.143 | 0.263 | 0.272 | 0.284 | 0.333 | 0.105 | 0.169 | 0.210 | 0.234 |
| LS | 0.016 | 0.071 | 0.150 | 0.147 | 0.446 | 0.390 | 0.375 | 0.319 | 0.209 | 0.250 | 0.249 | 0.245 |
| SB | 0.039 | 0.127 | 0.161 | 0.166 | 0.274 | 0.326 | 0.365 | 0.358 | 0.225 | 0.225 | 0.235 | 0.249 |
| QSB | 0.045 | 0.095 | 0.132 | 0.166 | 0.408 | 0.435 | 0.349 | 0.291 | 0.123 | 0.206 | 0.233 | 0.248 |

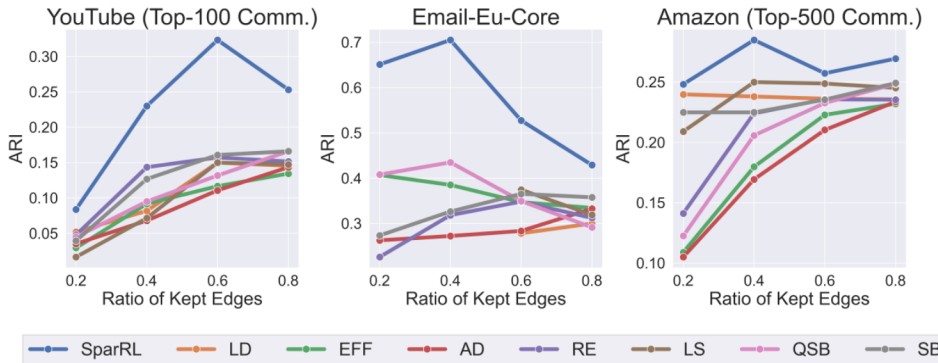

Figure 4: Community structure preservation measures over ARI score. SparRL outperforms all other methods on all cases.

where $\rho_{G'}$ is the Spearman's $\rho$ for the sparsified graph $G$ and $\rho_G$ is the Spearman's $\rho$ for the original graph $G$. The results in Figure 3 and Table 2 show that SparRL outperforms all other methods on the Twitter, Facebook, and Amazon (Top-500 Comm.) graphs.

**Community Structure Preservation**. We use the Adjusted Rand Index (ARI) (Hubert & Arabie, 1985) to measure the effectiveness of SparRL on preserving the community structure of a graph by comparing non-overlapping ground truth communities to those found using the Louvian method (Blondel et al., 2008) at multiple edge-kept ratios. We define the reward function as the different between ARI scores for $G$ and $G'$:

$$r_{com} = ARI(G') - ARI(G) + r_{label}, \tag{2}$$

where $ARI(G')$ is the Louvian ARI score on $G'$ and $ARI(G)$ is the original Louvian ARI score on $G$. The other reward, $r_{label}$, is defined as:

$$r_{label} = \begin{cases} 1 & l_{a_t^0} == l_{a_t^1} \\ -1 & else \end{cases}, \tag{3}$$

where $l_{a_t^0}$ is the label of the source node pruned at timestep t and $l_{a_t^1}$ is the label of the destination node. We add this auxiliary reward to encourage the agent to not prune an edge if its two nodes belong to the same community.

The results in Figure 4 and Table 3 show that SparRL consistently outperforms other methods on the YouTube (Top-100 Comm.), Email, and Amazon (Top-500 Comm.) graphs.

**Shortest Path Distance Preservation**. To test the ability of various sparsification methods in preserving the pairwise shortest path distance, We define the reward function for single-pair shortest path (SPSP) as follows:

Table 4: Comparison of SPSP preservation over the average increase of distance, where $r$ is the edge kept ratio.

| | Citeseer | | | | Email-Eu-Core | | | | Amazon (Top-500 Comm.) | | | |
|---|---|---|---|---|---|---|---|---|---|---|---|---|
| Method | $r$=0.2 | $r$=0.4 | $r$=0.6 | $r$=0.8 | $r$=0.2 | $r$=0.4 | $r$=0.6 | $r$=0.8 | $r$=0.2 | $r$=0.4 | $r$=0.6 | $r$=0.8 |
| SparRL | **2773** | **2473** | **338** | **15** | **0.326** | **0.133** | **0.07** | **0.021** | **281** | **106** | **0.518** | **0.24** |
| LD | 2896 | 2898 | 2899 | 730 | 0.355 | 0.16 | 0.078 | 0.033 | 299 | 121 | 77 | 9 |
| RE | 3233 | 2997 | 2047 | 1081 | 309 | 170 | 96 | 41 | 2930 | 1077 | 364 | 117 |
| EFF | 3064 | 2645 | 1896 | 696 | 173 | 56 | 30 | 18 | 2042 | 1126 | 211 | 56 |
| AD | 3252 | 3237 | 3165 | 2995 | 728 | 382 | 240 | 106 | 3803 | 3188 | 2460 | 1396 |
| LS | 3230 | 3232 | 3204 | 1191 | 107 | 54 | 32 | 31 | 2537 | 813 | 519 | 400 |
| SB | 3252 | 3192 | 3192 | 3191 | 891 | 531 | 321 | 250 | 2088 | 2084 | 1865 | 1400 |
| QSB | 3250 | 3192 | 3193 | 3193 | 970 | 465 | 364 | 292 | 3783 | 3022 | 2356 | 1319 |

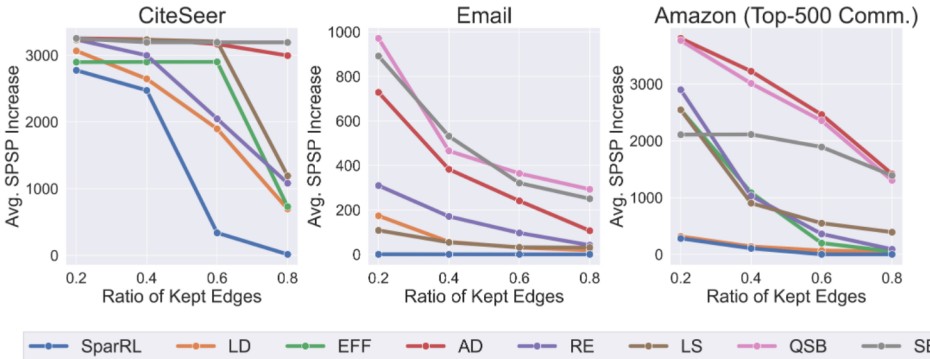

Figure 5: Shortest Path Distance Preservation measured by the average increase of distance between 8196 randomly selected pairs. In this figure, as SparRL's line is below all the others, it increases the SPSP distance the least and thus outperforms all other methods on all cases.

$$r_{spsp} = \frac{1}{|P|} \sum_{(u,v) \in P} dist(u,v)_{G'} - dist(u,v)_G, \qquad (4)$$

where $P$ is the set of SPSP pairs, and $dist(u,v)_{G'}$ is the SPSP distance between $u$ and $v$ in the sparsified graph $G'$, and $dist(u,v)_G$ is the SPSP distance between node $u$ and node $v$ in the original graph $G$. In the case where $v$ becomes unreachable from $u$ in $G'$, we set this difference equal to $|V|$, as this is greater than the maximum path length by 1.

During training, the set of SPSPs $P$ is created at each timestep before an edge is pruned. When the model chooses an edge to prune $a_t$, we sample random nodes and compute their shortest paths from the source node and destination node of $a_t$. This design is to leverage the optimal substructure property of shortest paths. That is, the only way a SPSP, for example from $u$ to $v$, will be affected by pruning edge $a_t$ is if it is contained in that path. This is because the SPSP between $u$ and $v$ must be composed of the SPSP from $u$ to the source node in $a_t$ and the SPSP from the destination node in $a_t$ to $v$. Thus, when $a_t$ is pruned, a new SPSP must be generated between $u$ and $v$ that does not contain the edge $a_t$.

During test time, however, we run a maximum of 8196 randomly sampled Single-Pair Shortest Path (SPSP) queries over the graph and keep them fixed for the entire episode. The results are given in Figure 5 and in Table 4 which show that SparRL consistently outperforms other methods on the CiteSeer, Email, and Amazon (Top-500 Comm.) graphs.

**Comparing SparRL and $t$-Spanner**. As $t$-spanner provides a way to sparsify a graph while preserving the geometric distance between a pair of nodes at most $t$ times of the original distance, we conduct an experiment study on comparing the performance of SparRL and a popular spanner algorithm given in (Baswana & Sen, 2007). Due to the fact that spanner algorithms cannot guarantee the number of edges to prune, we run the NetworkX (Hagberg et al., 2008) spanner implementation on various values of stretches and record its edge kept ratio. As it is an approximate algorithm, the

Table 5: SparRL compared against $t$-spanner for various stretch values $t$. ($x\%$: edge kept ratio)

| Method | $t$=3 (99.65%) | $t$=4 (99.63%) | $t$=8 (97.82%) | $t$=16 (93.74%) | $t = 32$ (90.78%) |
|---|---|---|---|---|---|
| t-spanner | 0.0082 | 0.0054 | 0.0405 | 0.1187 | 0.1911 |
| SparRL | **0.0031** | **0.0043** | **0.0350** | **0.0974** | **0.1820** |

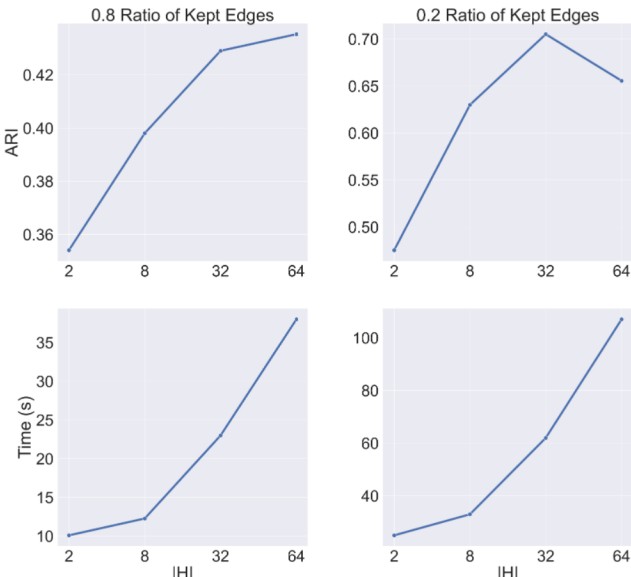

Figure 6: Subgraph length $|H|$ affect on model perfomance measured over Spearman's $\rho$ on the Email graph.

spanner algorithm produces a different sparsified graph each run. Therefore, we run the algorithm 16 times for each stretch value and compute the average number of edges and average performance over preserving randomly sampled SPSPs. We then run SparRL on the same average number of edges for each stretch value and display the $r_{spsp}$ (defined in Equation 4) results in Table 5 on the CiteSeer network. These results show that SparRL can outperform the approximate $t$-spanner algorithm over various stretch values.

## 5.3 Analysis of SparRL Components

**Impact of Subgraph Size**. At each timestep, SparRL inputs a subgraph which contains edges randomly sampled from $E_{G'_t}$. Due to the flexibility of our model architecture, we can have variable length subgraphs as input at test time. Thus, in Figure 6 we show results on applying the same trained model on varying subgraph lengths and measure their performance over Spearman's $\rho$ on the Email-Eu-Core graph. While increasing the subgraph length up to a certain point can improve model performance (e.g., for 0.8 edge-ratio $|H| = 64$ performance the best, even though SparRL was trained on $|H| = 32$), SparRL does increase the time to prune the edge. Thus, this is a time vs performance trade-off that can be adjusted accordingly based on the user's need.

## 6 Conclusion

In this work, we propose a general graph sparsification framework based on deep reinforcement learning, namely SparRL. SparRL can overcome the limitations of existing sparsification methods with relatively low computation complexity and the flexibility to adapt to a wide range of sparsification objectives. We evaluate SparRL using various experiments on many real-world datasets and representative graph metrics. The results show that SparRL is effective and generalizable in producing high-quality sparsified graphs. Further analysis on the components of SparRL have validated our design rationale. In the future, we would like to extend SparRL to the dynamic graph setting, and investigate how SparRL could help improve the performance of graph learning tasks, e.g., link prediction, label classification, and other graph related workloads.

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

# A  APPENDIX

## A.1  BASELINE METHODS

We use the following graph sparsification methods as baselines for our experiments. The implementations are from Networkit (Staudt et al., 2015) with their default settings.

- **Random Edge (RE)**: RE randomly prunes a given percentage of edges.
- **Local Degree (LD)** (Hamann et al., 2016): For each node $v \in V$, the edges in the top $\lfloor \deg(v)^\alpha \rfloor$ are kept in $G'$, where $\alpha \in [0, 1]$.
- **Edge Forest Fire (EFF)** (Hamann et al., 2016): Based on the Forest Fire node sampling algorithm (Leskovec & Faloutsos, 2006), a fire is started at a random node and burns approximately $p/(1-p)$ neighbor, where $p$ is the probability threshold of burning a neighbor. Any burnt neighbors are added to a queue to also have a fire started on them. It prunes edges based on the number of times each edge was visited.
- **Algebraic Distance (AD)** (Chen & Safro, 2011): Based on random walk distance, the algebraic distance $\alpha(u, v)$ between two nodes is low if there is a high probability that a random walk starting from $u$ will reach $v$ using a small number of hops. It uses $1 - \alpha(u, v)$ as the edge score so that short-range edges are considered important.
- **L-Spar (LS)** (Satuluri et al., 2011b): LS applies the Jaccard similarity function to nodes $u$ and $v$'s adjacency lists to determine the score of edge $(u, v)$. It then ranks edges locally (w.r.t each node) and prune edges according to their ranks.
- **Simmelian Backbone (SB)** (Nick et al., 2013): SB measures each edge $(u, v)$'s Simmelianness weight via the shared neighbors of $u$ and $v$. Then, for each $u$, it ranks its neighbors w.r.t the edge weights in descending order. During sparsification, SB will prune each node's lower ranked edges according to a given edge-prune ratio.
- **Quadrilateral Simmelian Backbone (QSB)** (Nocaj et al., 2014): QSB measures each edge $(u, v)$'s Simmeliannness weight via the shared quadrangles of $u$ and $v$. Then, it follows the same pruning strategy as that of SB.

