# OpenReview forum: "SparRL: Graph Sparsification via Deep Reinforcement Learning"
_ICLR.cc/2022/Conference — ICLR 2022 Submitted_

### Official Review · Reviewer_zwSi · 2021-10-30

**Correctness:** 3
**Technical Novelty And Significance:** 3
**Empirical Novelty And Significance:** 2
**Recommendation:** 3
**Confidence:** 4

**Main Review:**

This paper is overall well-rounded and introduces a solid, novel technique for a problem which (to paraphrase the authors) has received surprisingly little attention in the machine learning community. This paper addresses graph sparsification as the primary problem of interest, which is in contrast to earlier works (e.g. GDPNet and NeuralSparse, cited in the paper) which successfully use graph sparsification as „means to an end“ for solving a downstream task more efficiently. The experimental evaluation does a great job at demonstrating superior performance to a wide range of baselines on a wide range of graph datasets.

While I agree that using graph neural networks (GNNs) and reinforcement learning (RL) is novel for the particular task of graph sparsification as considered in this work, I wonder about the significance of this problem, given that other methods already successfully address graph sparsification to directly solve some kind of downstream task (e.g. GDPNet or NeuralSparse). I‘d recommend making a stronger argument for why this particular graph sparsification problem is important and why one cannot (or would not be advised to) directly optimize for the downstream task of interest. In its current form, I am unsure about the significance of this work — especially since an RL-based objective seems to be much more inefficient to train than the end-to-end differentiable graph sparsification technique (for a particular downstream task) introduced in NeuralSparse.

While the paper is overall well-written, it requires some further improvements in clarity in order to meet the bar for acceptance. For example, the index „n“ in Q(s_t, a_n) on page 4 is never defined. The overall description of the model architecture and the information flow through the model/policy in section 4.3 is difficult to parse, and would benefit from a more explicit definition using mathematical notation and/or a model diagram that visualizes the technique on an example sub-graph. The visualization in figure 2 does not help much in understanding the information flow through the graph. I am also not sure how GAT is being applied on directed graphs (since it is a method for undirected graphs). The notation in Eq. 1, page 5 is also unclear: f_SparRL is called a Q-value function but seems to correspond to a list(?) of multiple individual Q-value functions. It is not clear what the notation „Q(s_t, a_0), …, Q(s+t, a_|H|)“ represents— a vector-valued output of dimension |H|? In algorithm 1, it is also unclear to me why T_p is chosen to be the minimum of |E|*p and |E|-T.

Experimentally, I think the paper could be significantly improved by:
1) considering a downstream task of interest (e.g. node classification, or one of the tasks mentioned in the introduction) and showing that the proposed sparsification technique is competitive with end-to-end techniques such as NeuralSparse
2) ablating the choice of the GNN architecture: is attention necessary (e.g. what happens if we use a GCN [Kipf&Welling, ICLR2017] instead of GAT)? Is a GNN even necessary or does a single edge-based embedding function suffice?
3) reporting variance in the results over multiple seeds. This is absolutely crucial for a technique that involves multiple levels of sampling and a reinforcement learning target.

Lastly, it should also be discussed that the presented technique has to be trained and tested on the same graph and would not generalize if new nodes were added to a graph (since embeddings are learned for every single node).


**Summary Of The Paper:**

This paper introduces a method for graph sparsification using graph neural networks and reinforcement learning. The goal for graph sparsification considered in this work is to find edges so that certain global graph statistics are preserved upon removal of these edges from the graph. The authors use a single layer of a graph attention network to encode local graph structure and learn to predict an action-value function, following the framework of Q-learning, where actions correspond to the removal of a particular edge. The method is compared against several baseline heuristics and compares favorably on several small- to medium-scale network datasets (without node or edge features) in terms of preservation of graph statistics such as PageRank.

**Summary Of The Review:**

In summary, while I think that this paper is well-rounded and introduces a novel technique that outperforms a wide range of baselines, I am unsure about the significance of the considered problem. The paper has lots of headroom for improvement in terms of clarity and in terms of the experimental evaluation, which would help clarify its significance, the reliability of the results, and the necessity of core architecture choices (by adding additional ablation studies). Overall, I think that this paper is not ready for publication in its current form, but I am positive that many of the described issues can be addressed in a (potentially major) revision.

---

> ### Author Response · Authors · 2021-11-20
> **Thank you for your feedback**
>
> > I‘d recommend making a stronger argument for why this particular graph sparsification problem is important and why one cannot (or would not be advised to) directly optimize for the downstream task of interest.
>
> While SparRL is not designed specifically to solve some of the downstream tasks NeuralSparse is interested in, conversely NeuralSparse is also not fit to solve some of the problems we are interested in. For example, how can approaches like NeuralSparse solve the problem of reducing the edges of the graph while preserving the page rank score? Thus, our approach is solving problems that are not directly addressed by these methods and believe it is better left to future work
>
> > For example, the index „n“ in Q(s_t, a_n) on page 4 is never defined
>
> That is an error, we meant to put |H| instead of n. We can fix this in the revision.
>
> > The overall description of the model architecture and the information flow through the model/policy in section 4.3 is difficult to parse, and would benefit from a more explicit definition using mathematical notation .... (since it is a method for undirected graphs).
>
> We can add more explicit definitions in the revision
>
> >The notation in Eq. 1, page 5 is also unclear: f_SparRL is called a Q-value function but seems to correspond to a list(?) of multiple individual Q-value functions. It is not clear what the notation „Q(s_t, a_0), …, Q(s+t, a_|H|)“ represents— a vector-valued output of dimension |H|?
>
> The Q-value function maps state-action pairs to a scalar value. In SparRL, the model is run over each edge in the subgraph H in parallel separately. Thus, this creates the sequential Q-value function outputs. We can perhaps make this notation more clear in the revision though.
>
> >  In algorithm 1, it is also unclear to me why T_p is chosen to be the minimum of |E|*p and |E|-T.
>
> If |E| = 8 and T = 8, then |E| -T = 0, so we don't want to pre-prune more edges than we seek to prune in an episode.
>
> > considering a downstream task of interest (e.g. node classification, or one of the tasks mentioned in the introduction) and showing that the proposed sparsification technique is competitive with end-to-end techniques such as NeuralSparse
>
> The goal of this framework is to develop a general sparsification method that can solve many of the traditional edge sparsification problems. We considered testing against approaches like NeuralSparse, but we believe this is better left to future work. We did not look towards representational learning in this paper, because this would require adding further augmentations to the framework (e.g., setting up training a classification model that SparRL learns from, setting up a custom feedback loop, ect.), which we believe is out of scope for this paper.
>
> > ablating the choice of the GNN architecture: is attention necessary
>
> We can add a few extra experiments in the revision to test against other architectures; however, this may be something that would have to be included in the appendix as we already are tight in space with the current experiments we already perform in the paper.
>
> >   Reporting variance in the results over multiple seeds. This is absolutely crucial for a technique that involves multiple levels of sampling and a reinforcement learning target.
>
> We can include this in the revision.

---

> > ### Comment · Reviewer_zwSi · 2021-11-22
> > **Re: Thank you for your feedback**
> >
> > Thank you for your detailed response. I am looking forward to reading the revision of your paper. Regarding ablation studies: I think it is perfectly fine to defer these to the appendix, unless they carry a core message that is important to the contributions of the paper.

---

> ### Author Response · Authors · 2021-11-20
> **Thank you for your feedback 2**
>
> > .what happens if we use a GCN instead of GAT
>
> I forgot to add, the reason we used GAT as opposed to GCN is that GAT is invariant to the number of nodes in the 1-hop neighborhood of a node. Therefore, using a GCN isn't really applicable to our approach (unless we say arbitrarily sample a fraction of the 1-hop neighborhood of node, which would require an ablation study of its own), so we believe it's outside the scope of this project.

---

> > ### Comment · Reviewer_zwSi · 2021-11-22
> > **Re: Thank you for your feedback 2**
> >
> > Thank you for your response. This makes sense. In this case I would recommend comparing against a commonly used GCN variant with mean aggregation, i.e. simply dividing by the node degree of the receiving node as opposed to the geometric mean of the node degrees of both nodes along an edge. This is done for example in this PyTorch reference implementation of GCNs: https://github.com/tkipf/pygcn

---

### Official Review · Reviewer_na7w · 2021-11-01

**Correctness:** 4
**Technical Novelty And Significance:** 3
**Empirical Novelty And Significance:** 2
**Recommendation:** 6
**Confidence:** 4

**Details Of Ethics Concerns:**

No concern

**Main Review:**

Currently, learning based graph sparsification is still beyond the scope of this study area. The paper provides some new insights to graph sparsification and takes an initial step towards learning-based graph sparsification. Apparently, learning-based graph sparsification should be a promising direction for graph representation learning and graph neural network, where scalability is a bottleneck.

1. The work is a novel application of deep reinforcement learning on graph sparsification and technically sound from the perspective of graph sparsification.

2. Besides, the paper conducted a relatively thorough experiment to demonstrate the effectiveness and efficiency of the proposed SparRL framework. I like the empirical analysis of SparRL components, where gives different pruning strategy and expert control mechanisms are validated.

A main constraint from my perspective:

The empirical validation (experiment section) was only performed on some toy datasets, where the largest dataset is Citeseer containing 3312 nodes and 4536 edges. The empirical validation don't conform to the motivation of graph sparsification and triggers a doubt on SparRL’s scalability on massive graphs, since graph sparsification usually servers as an approximation technique for massive graphs. Though authors claimed that the effectiveness of SparRL is independent from the size of graphs, it would be more convincing to demonstrate it on massive graphs, where #nodes should be more than millions.

**Summary Of The Paper:**

This paper proposes a learning-based graph sparsification framework (SparRL) using reinforcement learning and imitation learning. The proposed framework is task-adaptive and performs well under several evaluation metrics like PageRank, community structure, and pairwise shortest-path distance.

**Summary Of The Review:**

The paper provides a novel RL framework for learning based graph sparsification. The work is technically sound and well-organized in writing, However, empirical validation on large-scale graphs is missing, which prevents me from an acceptance decision.

---

> ### Author Response · Authors · 2021-11-20
> **Thank you for your feedback**
>
> >  The empirical validation (experiment section) was only performed on some toy datasets, where the largest dataset is Citeseer containing 3312 nodes and 4536 edge
>
> We *did* test SparRL on bigger graphs, with the largest being Twitter with 81,306 nodes and 1,768,149 edges.
>
> > Besides, the paper conducted a relatively thorough experiment to demonstrate the effectiveness and efficiency of the proposed SparRL framework. I like the empirical analysis of SparRL components, where gives different pruning strategy and expert control mechanisms are validated
>
> So, I think this review is pertaining to one of our old papers submitted to a different conference. We did not have the expert control mechanisms in this paper.

---

> > ### Comment · Reviewer_na7w · 2021-11-23
> > **Thanks you for the reponse**
> >
> > Thanks for the response of my questions. I confirm that some of my comments have already been addressed. I hereby increase my score.

---

### Official Review · Reviewer_AdUd · 2021-11-01

**Correctness:** 2
**Technical Novelty And Significance:** 3
**Empirical Novelty And Significance:** 2
**Recommendation:** 3
**Confidence:** 4

**Main Review:**

Major comments:
- As the submitted paper is application-oriented, the authors should explain more of implementation details. Elaboration of certain details are indispensable for the clarity and novelty of this work. Here are some examples:
     -  page 4, section 4.2: "Each time step ... ... to avoid sparse rewards". Under what circumstances do sparse reward appear? Is not the experiment involved episodic? If the experiment is episodic, then why does evaluating reward at each time step contribute to the purpose of "avoiding sparse rewards"?
     -  page 4, section 4.3: what particular neural network structure the authors use to represent sparsification policy?
     -  page 8, equation (4): how do authors address the cases when $\mathrm{dist}(u,v)_{\mathrm{G}'} = \infty$ due to the removal of an edge

-  What is $|H|$ in algorithm 1? This particular notation is not defined.

- One main advantage of the proposed method over $t-$spanner is the guarantee of edge reduction ratio in the output graph. I find the description of the edge sparsity control confusing in algorithm 1 on page 4. Say, the edge reduction ratio is $p=0.5$ and the sampling ratio is correspondingly $1-p$, why in algorithm 1 that $p$ is "randomly sampled"? Should not this be an input quantity?

  Considering the size of network studied in this paper, the motivation of setting $T_{\text{max}} = 8$ is not clear to me.

- Table 5 on page 9: Now that ($x%$%) means edge kept ratio, why when comparing SparRL with $t-$spanner, the edge-kept ratio is set so high? On which dataset is this comparison carried out?

- A few places where I find the description in accurate:
     - page 6, definition of $r_{\textrm{PR}}$: should not the expression in this definition the loss instead of the "reward"?
     - page 8, equation (4), definition of $r_{\textrm{SPSP}}$: should not the expression in this equation the loss for single-pair shortest distance preservation instead of 'reward'?

Minor comments:
  I recommend the author revise some notations (for example, maybe not use '$*$' to mean multiplication) and grammar errors in the writeup.

**Summary Of The Paper:**

This paper discusses the protocol "SparRL", which is a reinforcement learning driven technique for sparsifying graphs to a given edge reduction ratio while preserving the graph structure.

In this RL-driven graph sparsifying protocol, the authors describe the edge reduction process as a Markov dynamic process and leverage the double deep Q-learning technique to represent the value function, thus conducting policy learning.

For evaluating the effectiveness of graph sparsification techniques, the authors investigate aspects of PageRank preservation, community structure preservation and shortest path distance preservation. The authors benchmark the proposed RL-driven methods with several methods in literature.

**Summary Of The Review:**

The paper is not well written in terms of the lack of clarity in stating technical details and the unconvincing presentation of simulation results. Please refer to Main Review session for point-by-point questions.

The paper is not ready to be published.

---

> ### Author Response · Authors · 2021-11-20
> **Thank you for your review**
>
> > page 4, section 4.2: "Each time step ... ... to avoid sparse rewards". Under what circumstances do sparse reward appear? Is not the experiment involved episodic? If the experiment is episodic, then why does evaluating reward at each time step contribute to the purpose of "avoiding sparse rewards"?
>
> I think you may be confused about sparse rewards, because it has nothing to do with the environment being episodic, so let me give you a quick example. Say we have an environment with a maximum of T = 8 timesteps. Say we only assign the reward of 1 at the last timestep, then the rewards would be [0, 0, 0, 0, 0, 0, 0, 1]. As most the values in this vector are 0, it is sparse. Now if instead we compute the reward every timestep, then it could be [0.125, 0.125, 0.125, 0.125, 0.125, 0.125, 0.125, 0.125], which is not sparse.
>
> > page 4, section 4.3: what particular neural network structure the authors use to represent sparsification policy?
>
> An overview is given in Figure 2 and it is described in depth in Section 4.3
>
> > page 8, equation (4): how do authors address the cases when dist(u, v) =  ∞ due to the removal of an edge
>
> This is discussed in Section 5.2 under **Shortest Path Distance Preservation** where we say, "In the case where v becomes unreachable from u in G′, we set this distance equal to|V|,as this is greater than the maximum path length by 1."
>
> > What is |H| in algorithm 1? This particular notation is not defined
>
> We did define this in Section 4.3 where say |H| is the subgraph length.
>
> > why in algorithm 1 that is "randomly sampled"? Should not this be an input quantity?
>
> This algorithm is what we do during training SparRL.
>
> > Considering the size of network studied in this paper, the motivation of setting Tmax = 8 is not clear to me.
>
> In Section 4.1, we discuss how we randomly initialize the initial state to remove the requirement of exploration. Thus, while training, we do not *need* to have longer episodes. This value is set to a higher value during evaluation.
>
> > Table 5 on page 9: Now that (%) means edge kept ratio, why when comparing SparRL with spanner, the edge-kept ratio is set so high? On which dataset is this comparison carried out?
>
> We say in Section 5.2 under **Comparing SparRL and t-Spanner** it is on the CiteSeer graph. The reason the edge-kept ratio is high is because the *t*-spanner algorithm does not have much freedom on the number of edges that should be kept in the graph. So, we tested on various *t*s for *t*-spanner and pruned the same amount of edges that it did.
>
> > page 6, definition of r_PR: should not the expression in this definition the loss instead of the "reward"?
>
> No. This is a reward not a loss function as we are using reinforcement learning to train SparRL
>
> > page 8, equation (4), definition of : should not the expression in this equation the loss for single-pair shortest distance preservation instead of 'reward'
>
> Again, this is a reward not a loss as we are using reinforcement learning to train SparRL

---

### Official Review · Reviewer_eytn · 2021-11-06

**Correctness:** 4
**Technical Novelty And Significance:** 2
**Empirical Novelty And Significance:** 2
**Recommendation:** 3
**Confidence:** 4

**Main Review:**

Strengths:

- Graph sparsification is a useful technique to support efficient processing of big graphs for various applications that may depend on specific graph properties but may not require the presence of entire original graph structure. Further, the proposal of automated learning framework to perform sparsification aligns with recent advances in using reinforcement learning techniques for combinatorial optimization problems [1].  Hence this work will be of interest to the community.
- Compared to baselines that are designed to focus on preserving specific properties, the proposed approach is able to support preserving arbitrary properties as long as they can be encoded in the form of a reward function.
- The approach depends on the number of edges in the sampled subgraph of preprocessed sparsified graph, but does not depend on size of original graph which is a plus and may provide some gains over computationally expensive sampling methods
- Empirically, the approach is shown to perform well compared  to all baselines across all settings
- Authors present discussion on time vs performance tradeoff which is quite useful to discern some understanding on the time complexity.


Weaknesses:

- The overall approach of using reinforcement learning for optimization of transforming graphs or obtaining a substructure of graph while preserving some properties is not entirely novel. The authors are encouraged to refer to the approaches in [1] and compare and contrast how their approach is novel compared to other Q learning based approaches for performing optimization problems on graphs
- The description of overall approach is not clear and misses some details due to which it is hard to discern whether the performance of the method is due to the pre-processing and other sparsification parameter choices (size of H, T_max, etc) or outcome of learning itself. Better description, ablation studies and hyper-parameter sensitivity analysis  can help with this. Also, the overall approach of pre-pruning and sampling subgraphs from pre-pruned graph to induce POMDP is very adhoc and the authors need to describe the rationale behind these choices and analyze how it affects the performance.
- Following are some specific questions related to that:
1. The effect of pre-pruning the graph is not clear and needs to be analyzed more. Also, as T is taken to be a very small number during training, it seems like the pre-pruning is almost always dominated by p and a significant number of edges are already pruned at the start of episode. Doesn’t this lead to situation where some edges are never part of the pre-pruned graph and hence not seen during training? Why is small T useful?
2. How does node degree input contribute to the overall performance. Given that the three metric used to compare the performance are directly relevant to the node degree, is that most important factor for the performance?
3. Could the authors elaborate more on what they mean by “allows agent to start in any part of state space without pruning T_p edges”. Doesn’t the approach pre-prune T_p edges?


- The baselines are not particularly optimized for preserving the graph properties on which they are compared which is clear disadvantage for them.
- It is important to compare with [2] and [3] even when they do not particularly perform edge removal. [2] performs representation learning but it has ability to sample sparse graph and also displays them in that paper. Comparing the performance of current approach agains such graph is important. For [3], while it allows edge generation, the overall performance need to be compared with this method as baseline as it will implicitly remove some edges and add others, thereby potentially better capturing the properties while reducing overall number of edges significantly.
- It is important to show visualization of the original graph and sparse graphs (may be a useful subgraph) to illustrate property preservation such as communities.
- From the tables, it is seen that some edge kept ratios (e.g. in Email datatset) have higher variance in performance across ratios compared to other settings. Can the authors elaborate on why this is the case?
- As the experiments were done for 8 runs, it is useful to plot the error bars/ deviation across runs to get clear picture of performance.

Other comments:

- Two major claims of authors include better computational complexity than sampling methods and flexibility. For flexibility, I believe the authors limit to claim to the extent of being able to use same learning procedure with different reward function representing different properties. However, one would need to retrain from scratch for preserving each different property. Is that correct? For time complexity, while the authors provide some insights into the runtime of their own method, they need to provide rigorous comparison with complexity of other methods to support their claim.

[1] Reinforcement Learning for Combinatorial Optimization: A Survey, Mazyavkina et. al., 2020

[2] Robust Graph Representation Learning via Neural Sparsification, Zheng et. al. ICML 2020

[3] Graph Sparsification with Generative Adversarial Network, Wu et. al. 2020


**Summary Of The Paper:**

The paper focuses on the problem of graph sparsification - a problem of approximating an arbitrary graph with a sparse graph (specifically the one produced only by the  removal of edges in the context of this paper) while retaining the desired structural properties of the graph. Compared to previously proposed classical heuristics approaches or task specific learning approaches, the authors propose an automated learning framework for graph sparsification by considering edge removal as a sequential decision making process and solve it using Q learning. The learning procedure employs a preprocessing to sample initial state as a randomly sparsified subgraph to allow the agent to start at random parts of the state space. The state space is a subgraph and action is an edge that is to be removed. Within each  step of an episode, the action computation uses partial state information in the form of  sampled edges of the initial state graph and combination of degree of nodes and edge kept ratio (number of edges in sparse graph compared to original graph). Desirable properties that need to be retained are encoded in the form of reward function whose definition varies based on graph properties (page rank, shortest path and community structure in this paper). The architecture consists of node and edge encoders based on Graph Attention networks followed by a MLP based value function. Empirical evaluations are done on 6 different graphs with different characteristics and compared against representative classical baseline approaches for sparsification. Authors report performance based on correlation between the topological structure of sparsified and original graph with respect to the three properties mentioned above. The authors demonstrate that the proposed approach outperforms all baselines across different edge kept ratios and further discuss time vs performance tradeoff.

**Summary Of The Review:**

The proposed work tackles an interesting problem and provides a useful technique to perform automated sparsification of graphs based on user required edge kept ratio and property specific objective. However, the overall  approach is not novel from previous works in reinforcement learning on graphs and the differing design choices are not discussed or analyzed adequately. Further some important comparisons are missing and it is not clear if this method can scale in performance critical applications as it is shown by the authors that time vs performance tradeoff is significant depending on size of sampled subgraph. Hence, this work is not ready for publication in its current form.

---

> ### Author Response · Authors · 2021-11-20
> **Thank you for your feedback (1/2)**
>
> > The effect of pre-pruning the graph is not clear and needs to be analyzed more. Also, as T is taken to be a very small number during training, it seems like the pre-pruning is almost always dominated by p and a significant number of edges are already pruned at the start of episode. Doesn’t this lead to situation where some edges are never part of the pre-pruned graph and hence not seen during training? Why is small T useful?
>
> As we discuss in Section 4.1, we pre-prune the graph so that we can remove the requirement of exploration for the agent. We a pre-pruning *before every episode* during training, so as long we train on a sufficient number of episodes, it should visit every edge. Although it really only has to visit every node in the graph as the edge embeddings are created from the node embeddings.
>
> > How does node degree input contribute to the overall performance. Given that the three metric used to compare the performance are directly relevant to the node degree, is that most important factor for the performance?
>
> As this is POMDP and the degrees of the nodes change everytime an edge is pruned, this is important information that should be given to the graph. For example, say we know that a node only has a degree of 1, then prunning it's edge would disconnect it from this graph. If this information is not given to SparRL, it would have no way of knowing this important information.
>
> > Why is small T useful?
>
> As the state space of this environment includes all G' = (E', V), where E' is a subset of E, the number total states of the graph is rather large. Thus, training for too long on one randomly sampled G' could make it overfit to that initialization. However, most importantly, we are doing pre-prunning to *remove* the requirement complex exploration mechanisms from the agent. If T was set to a larger value, we would have to focus more on exploration, would would defeat part of the original purpose of performing pre-prunning.
>
> > Could the authors elaborate more on what they mean by “allows agent to start in any part of state space without pruning T_p edges”. Doesn’t the approach pre-prune T_p edges?
>
> We only pre-prune edges during training, as said in Section 4.1. Let me give an example to make this more clear though. Say we have a graph where |E| = 10,000 and don't use pre-prunning, then to visit the state(s) of the graph that have |E|=3000, we would have to first prune 7000 edges in the episode, which would greatly increase the total training time. Instead, we pre-prune 7000 edges and set the initial state to |E| = 3000. This allows SparRL to know the best thing to do at each state of the graph, without requiring a vast number of timesteps and heuristic exploration to reach them.
>
> > The baselines are not particularly optimized for preserving the graph properties on which they are compared which is clear disadvantage for them.
>
> As we are developing a general sparsification method, we tested against general sparsification methods. We did test against the *t*-spanner algorithm, which is said to optimally preserve the shortest paths. However, since its implemented using an approximate version, we were able to outperform it. Furthermore, we used the same baselines as in [3] and other traditional papers, so we did not view this to be a problem.
>
> > It is important to compare with [2] and [3] even when they do not particularly perform edge removal. [2] performs representation learning but it has ability to sample sparse graph and also displays them in that paper.
>
> We considered testing against [2], but we believe this is better left to future work. As our original goal of the SparRL framework was to compare against traditional edge sparsification algorithms, we did not look towards representational learning in this paper. This would require adding further augmentations to the framework (e.g., setting up training a classification model that SparRL learns from, setting up a custom feedback loop, ect.), which we believe is out of scope for this paper.

---

> > ### Author Response · Authors · 2021-11-20
> > **Thank you for your feedback (2/2)**
> >
> > > For [3], while it allows edge generation, the overall performance need to be compared with this method as baseline as it will implicitly remove some edges and add others, thereby potentially better capturing the properties while reducing overall number of edges significantly.
> >
> > As [3] adds edges to the original graph, we believe this is not a fair comparison and not in line with what the current objective of our work is focused on. We focus on *only* removing edges from the graph and compared to methods and metrics that focus exactly on that. If we wanted to do a fair comparison, we would give SparRL the ability to add as well as remove edges. Thus, this is something that we believe is better left to future work. Another important fact to note is for community detection, if you are allowed to remove *and* add edges to the graphs, the algorithm could quickly learn to remove edges that are not between communities and add edges between communities, which seems like a trivial solution that does not require a learning based approach to perform.
> >
> > > It is important to show visualization of the original graph and sparse graphs (may be a useful subgraph) to illustrate property preservation such as communities.
> >
> > We did provide a toy example in Figure 1, however, as the test graphs are rather large, we found that adding these figures did not really help convey much information as it gets rather cluttered (e.g., as |V| and |E| increases). Plus we found that other work did not include these figures for all the test graph, so we did not.
> >
> > >  From the tables, it is seen that some edge kept ratios (e.g. in Email datatset) have higher variance in performance across ratios compared to other settings. Can the authors elaborate on why this is the case?
> >
> > We believe this a a factor of the graph itself rather than SparRL (as other algorithms seem to have a similar pattern on the same graph), and thus is not an important thing to discuss.
> >
> > > As the experiments were done for 8 runs, it is useful to plot the error bars/ deviation across runs to get clear picture of performance.
> >
> > This is something we can add in the revision.
> >
> > > However, one would need to retrain from scratch for preserving each different property. Is that correct?
> >
> > SparRL learns based on the reward function that is designed specifically for that objective. Having a more general reward function, that works for *all* objectives is not something so trivial, so we believe should be left to future work.

---

> > > ### Comment · Reviewer_eytn · 2021-11-30
> > > **Thank you for your response**
> > >
> > > I have read the responses carefully and I would like to retain my score. The authors have responded to all of my concerns, however, most of them are expressed as justifications of not doing what was requested instead of resolving the concern or performing the experiments. I believe that most of the comparisons and tasks (that author have classified as future work in their response) suggested here are useful and important to make the paper stronger. Given this, I do not think this paper is ready for acceptance in this conference.

---

### Decision · Program_Chairs · 2022-01-20

**Decision:**

Reject

**Comment:**

The paper presents an RL approach to the problem of graph sparsification. The reviewers expressed concerns about novelty, presentation, the correctness of some claims, and experimental validation. While the authors provided rebuttal and addressed some questions (leading to the increase of the score), some reviewers thought the authors focused on a justification of why suggested experiments were not done rather than doing them. We believe the paper in its current state is below the bar and recommend rejection.